# Participation and Outcomes among Disabled and Non-Disabled People in the Diabetes Pay-for-Performance Program

**DOI:** 10.3390/healthcare11202742

**Published:** 2023-10-16

**Authors:** Wei-Yin Kuo, Wen-Chen Tsai, Pei-Tseng Kung

**Affiliations:** 1Department of Health Services Administration, College of Public Health, China Medical University, Taichung 406040, Taiwan; u100050853@cmu.edu.tw (W.-Y.K.); wtsai@mail.cmu.edu.tw (W.-C.T.); 2Department of Medical Research, China Medical University Hospital, Taichung 404332, Taiwan; 3Department of Healthcare Administration, Asia University, Taichung 413305, Taiwan

**Keywords:** disabilities, diabetes, pay-for-performance program, dialysis, survival

## Abstract

Objectives: This study’s objectives were to compare the participation rates of people with and without disabilities who had type 2 diabetes in a diabetes pay-for-performance (DM P4P) program, as well as their care outcomes after participation. Methods: This was a retrospective cohort study. The data came from the disability registry file, cause of death file, and national health insurance research database of Taiwan. The subjects included patients newly diagnosed with type 2 diabetes between 2001 and 2013 who were followed up with until 2014 and categorized as disabled and non-disabled patients. The propensity score matching method was used to match the disabled with the non-disabled patients at a 1:1 ratio. Conditional logistic regression analysis was used to determine the odds ratio between the disabled and non-disabled patients who joined the P4P program. The Cox hazard model was used to compare the risk of dialysis and death between the disabled and non-disabled patients participating in the P4P program. Results: There were 110,645 disabled and 110,645 non-disabled individuals after matching. After controlling for confounding factors, it was found that the disabled individuals were significantly less likely (odds ratio = 0.89) to be enrolled in the P4P program than the non-disabled individuals. The risk of dialysis was 1.08 times higher for people with disabilities than those without, regardless of their participation in the P4P program. After enrollment in the P4P program, the risk of death for people with disabilities decreased from 1.32 to 1.16 times that of persons without disabilities. Among the people with disabilities, the risk of death for those enrolled in the P4P program was 0.41 times higher than that of those not enrolled. The risk of death was reduced to a greater extent for people with disabilities than for those without disabilities upon enrollment in the DM P4P program. Conclusion: People with disabilities are less likely to be enrolled in the P4P program in Taiwan and have unequal access to care. However, the P4P program was more effective at reducing mortality among people with disabilities than among those without.

## 1. Introduction

Approximately 537 million people worldwide have diabetes, with a prevalence rate of 9.8%, and 6.7 million deaths have been caused by diabetes [1]. Proper glycemic control can reduce the risk of neuropathy by 60%, retinopathy by 76%, and nephropathy by 35–56% [2]. The United States Renal Data System 2020 reported a 47.1% rate of hemodialysis among patients with kidney disease caused by diabetes [3], indicating that renal failure is one of the most serious complications of diabetes, which may eventually necessitate dialysis or kidney transplantation. The related literature indicates that the incidence of end-stage renal disease (ESRD) in diabetic patients is 3–12 times higher than that in non-diabetic patients [4,5].

People with disabilities are prone to having multiple chronic diseases and poor overall health [6]; therefore, they tend to utilize healthcare more and consequently incur greater costs [6,7,8]. However, the quality of care they receive and the prevention of complications (such as diabetes) are poor, meaning that they suffer from more severe disease and higher mortality compared to the general population.

Taiwan’s national health insurance covers 99.9% of the population [9]. The diabetes pay-for-performance plan (DM P4P) was implemented in November 2001. The DM P4P program provides complete services, including diagnosis, examination, health education, and follow-up, to reduce or delay the complications of diabetes [10]. Diabetic patients can only be enrolled if they have been diagnosed with diabetes mellitus (the first three codes of ICD9CM: 250) more than twice within 90 days by the same physician at the same institution or if they have been hospitalized for diabetes mellitus [10]. Physicians need to provide diabetes self-management education to patients enrolled in the P4P program and regularly follow up with them, at least three times a year, with biochemical tests measuring HbA1c, serum creatinine, and LDL cholesterol [10]. In addition to the regular physician consultation fees and management care fees, physicians who rank in the top 25% in terms of quality of care can receive additional case incentives [11]. The percentage of patients enrolled in the DM P4P program in 2013 was 35.1%, which increased to 51.3% in 2018 [12]. However, the situations of people with disabilities in the P4P program have not been explored.

Most studies have confirmed that the implementation of the P4P program improves the quality of care for diabetic patients while controlling medical costs [13,14]. An American study [15] examined diabetes disease management programs (DDMPs) and found that diabetic patients who joined the management programs had fewer hospitalization days and fewer expenses than those who did not. Their glycated hemoglobin tests, blood lipid tests, retinal tests, microprotein tests, and smoking habits were better than those of non-enrollees. Another British study also showed that the quality of medical care for diabetes has significantly improved since the pay-for-performance contract was introduced [16]. Research in Taiwan has also shown that the diabetes pay-for-performance program can improve the quality of care for diabetic patients and control medical costs at the same time [17]. In addition, enrollment in diabetes pay-for-performance programs can improve the continuity of care and reduce hospitalization and emergency room utilization for diabetes-related diseases [18,19]. However, there is no research on the care outcomes of people with disabilities participating in P4P programs.

Disabled people can apply for a disability certificate for obtaining social welfare and medical subsidies from the Ministry of the Interior (MOI) after being diagnosed by a qualified specialist from a public hospital [20]. According to the Ministry of Health and Welfare (MOHW) statistics from 2023, there were 1,196,654 disabled people in Taiwan in 2022, accounting for approximately 5.14% of the total population [21]. People with disabilities have a higher risk of developing diabetes than the general population [22,23,24]. The age-standardized prevalence of diabetes among the disabled is 15.8%, which is much higher than the age-standardized prevalence of 7.2% among all adults in the United States [25]. Furthermore, people with intellectual disabilities have a higher risk of developing diabetes than those with non-intellectual disabilities, with prevalence ranging from 7.1% to 14% [22,23].

Many adults with disabilities have more sedentary lifestyles [26], low levels of physical activity [27], and high-fat diets [28], which may lead to obesity and, consequently, the development of diabetes. Another study confirmed that the use of antipsychotic drugs (clozapine) is positively related to obesity (BMI > 30) [29], resulting in increased risks of dyslipidemia and diabetes [30,31]. The poor overall health of people with disabilities puts them at risk for more severe comorbidities and higher mortality rates compared to the general population [7,32,33,34].

Few studies have examined the enrollment rates and care outcomes for people with disabilities. Previous studies have shown that patients with more severe comorbidities [35,36,37], more severe diabetes complications, and more complex chronic illnesses [10,35,37] are less likely to enroll in the DM P4P program. This suggests that the Taiwan DM P4P program tends to screen patients and that diabetic patients enrolled in the program have significantly less severe conditions than those not enrolled. Thus, this study aimed to investigate the differences between disabled and non-disabled diabetic patients enrolled in the DM P4P program and their associated factors. This study also aimed to analyze the differences in care outcomes, including the risks of dialysis and mortality, when disabled diabetic patients enrolled in the DM P4P program.

## 2. Materials and Methods

### 2.1. Data Sources and Participants

This is a retrospective cohort study. The data sources included the people registered as having disabilities by Taiwan’s MOI from 2000 to 2008, the National Health Insurance Research Database maintained by Taiwan’s Ministry of Health and Welfare, the Registry for Catastrophic Illness Patients, and cause of death data. The study population included patients who were newly diagnosed with type 2 diabetes between 1 January 2001 and 31 December 2013. The period from 2000 served as the wash-out period for newly diagnosed diabetics. With reference to previous related studies [19], in this study, diabetic patients were defined as those with three primary or secondary outpatient diagnoses of diabetes (ICD9: 250, A181) or one hospitalization within 365 consecutive days [36,38,39,40], using the date of the first diabetes visit as the date of confirmation (index date) while excluding gestational diabetes (ICD9CM: 648.0, 648.8) and type 1 diabetes (ICD9CM: 250.x1, 250.x3, 250.1x).

In this study, patients with diabetes were divided into those with disabilities and those without disabilities. The study subjects were defined as disabled according to data from the registry of people with disabilities from Taiwan’s MOI. Furthermore, the occurrence of dialysis or death was tracked up to 31 December 2014. The following were excluded: (1) those who had diabetes prior to acquiring a disability; (2) those who were in a vegetative state; (3) those with disabilities as well as congenital metabolic abnormalities related to metabolic diseases; and (4) those who had a kidney transplant or dialysis before developing diabetes.

Many studies on the DM P4P program have shown that the program tends to screen patients; in particular, those with a higher Charlson Comorbidity Index (CCI) and Diabetes Complications Severity Index (DCSI) scores were less likely to be enrolled in the P4P program [10,35,37]. The disabled people in the studies have poorer health than those without disabilities [6] and tend to have more severe CCI scores [7,34]. Therefore, to prevent the effects of the inherent differences between the disabled and non-disabled participants from influencing the study’s results, because our research was a national database analysis, the number of disabled people selected was 110,645. For avoidance, the total number of people after matching was too large, which would have easily led to statistically significant differences during analysis. Therefore, we used a 1:1 propensity score matching (PSM) method using the OneToManyMTCH macro proposed by Parsons, Ovation Research Group, Seattle, WA, USA [41]. Logistic regression analysis was used to calculate the propensity scores of disabled and non-disabled patients who enrolled in DM P4P program; then, we matched them in the hierarchical order of eight digits to one digit in the greedy algorithm (Figure 1).

Before 2012, persons with disabilities were evaluated by qualified medical specialists from public hospitals and their information was sent to the Social Bureau in each city for registration before being sent to the Ministry of the Interior. There are sixteen categories of disability and four levels of severity (mild, moderate, severe, and very severe). The data used in this study were from the Disability Registration Files, and include the type, severity, cause of disability, and date of disability.

To reduce the financial burden of long-term healthcare for people with serious illnesses, the Taiwan National Health Insurance Administration (NHI) provides financial exemptions for people with serious illnesses who meet the NHI’s definition of catastrophic illnesses as diagnosed by physicians, including cancer, stroke, dialysis, long-term respirator dependence, and another 26 categories. Dialysis patients are diagnosed by a nephrologist as requiring long-term dialysis due to kidney failure and are provided with indications for dialysis and laboratory data. The date of application (date of illness), types of major illness, and ICD-9-CM diagnosis are recorded in the Registry for Catastrophic Illness Patients database.

### 2.2. Description of Variables

Subject to the regulations of Taiwan’s DM P4P program and the previous related literature [19], those with a primary diagnosis of diabetes, outpatient case classification of “E1”, specific treatment item code of “E4”, or inpatient case category of “C” were defined as having enrolled in the DM P4P program, with the date of first report recorded as the index date.

Dialysis patients were defined as patients recorded as being on dialysis in the Catastrophic Illness database. If a patient died, the date of death was determined from the cause of death file. The rest of the variables and their definitions are listed below.

Disabled: Those who are registered as disabled, including their gender and age at the index date. CCI: The patient’s disease diagnosis codes within two years prior to the index date were used to calculate the CCI score based on the Charlson Comorbidity Index as modified by Dey et al. [42]. DCSI: In 2008, Young et al. classified diabetic complications into seven categories, with diagnoses of complications within two years before the index date [43]. Hypertension (ICD9CM: 402, 405) and hyperlipidemia (ICD9CM: 272.0–272.1): Two outpatient visits or one diagnosis due to hospitalization. Monthly salary: Salary income at the end of the index year. Level of urbanization: All townships were categorized according to seven levels of urbanization, the first of which was the most urbanized and the seventh of which was the least urbanized [44]. Characteristics of health providers: The primary healthcare provider was defined as the institution and physician who received the largest number of medical visits from diabetic patients (primary or secondary diagnosis of 250 or A181). Service volume of physicians: The annual service volume of the primary physician was ranked among the annual service volumes of all physicians who provided outpatient care for diabetic patients. The annual service volume of each primary care physician for diabetes was categorized as high service volume (>Q3), medium service volume (Q3 to Q1), or low service volume (<Q1). Level of primary care facilities: Divided into medical centers, regional hospitals, district hospitals, and primary care clinics. Ownership of institution: Categorized as public hospitals and non-public hospitals.

### 2.3. Statistical Analysis

Statistical analysis was conducted using SAS 9.4 (SAS Institute, Cary, NC, USA) software with a confidence level of α = 0.05 to determine the differences between the dependent variables (enrollment in the P4P program) according to the independent variables, including status (disability and non-disability), basic personal characteristics (gender and age), health status (CCI, DCSI, hypertension, and hyperlipidemia), economic factors (monthly salary), environmental factors (level of urbanization), and the characteristics of the primary health providers (service volume of the primary physician, level of primary care facility, and ownership).

A conditional logistic regression analysis was then conducted to examine the odds ratios between the disabled and the non-disabled and whether they enrolled in the P4P program or not, controlling for basic personal characteristics, health conditions, economic factors, environmental factors, and the characteristics of primary health providers.

Finally, we evaluated the effects of enrollment in the P4P program on the care outcomes of disabled and non-disabled people by examining the risk of dialysis and the risk of death. As the care outcomes of disabled and non-disabled patients after joining the P4P program may vary, the Mantel–Haenszel chi-square was used to examine these interactions. The Cox proportional hazard model (Cox PH model) was then used to control for relevant factors to examine the differences in the risk of dialysis and the risk of death between disabled and non-disabled patients who did or did not enroll in the P4P program.

## 3. Results

### 3.1. Results of Matching between Disabled and Non-Disabled Diabetic Patients

There was a total of 2,474,399 diabetic patients diagnosed between 2000 and 2013. After exclusion, 1,730,891 people from 2001 to 2013 remained, of whom 1,620,246 were non-disabled and 110,645 were disabled. The percentage of disabled males was 56.2%, with an average age of 63.3 years, whereas the proportion of non-disabled males was 53.4%, with an average age of 57.9 years. Gender, age, CCI, and DCSI significantly differed between disabled and non-disabled patients (*p* < 0.05). After 1:1 matching through PSM, there were 110,645 non-disabled patients (average age of 62.1 years) and 110,645 disabled patients (average age of 62.4 years) included, resulting in a total of 221,290 study participants (Table 1). There were no statistically significant differences in gender, age, CCI, or DCSI (*p* > 0.05).

### 3.2. Comparison of Enrollment in the P4P Program among Diabetic Patients with and without Disabilities

Among the 221,290 diabetic patients, 47,945 (21.7%) were enrolled in the P4P program and 173,345 (78.3%) were not enrolled (Table 2). A total of 22,254 (20.1%) of the disabled patients were enrolled in the P4P program, while 23.2% (25,691) of the non-disabled patients were enrolled in the P4P program. The percentage of people with disabilities who joined the P4P program was lower than that of those without disabilities, regardless of gender and age, and only 6.94% of people with disabilities aged 75 years and older enrolled in the P4P program. The average age of the disabled people who were enrolled in P4P was significantly lower. Only 10.8% of disabled people with CCI scores greater than three were enrolled in the P4P program, while 14.5% of non-disabled people were enrolled. Likewise, only 9.12% of disabled people with DCSI scores greater than three were enrolled in the P4P program, whereas this statistic for non-disabled people was 13.2%. As a result, in the context of the same personal characteristics, health status, and socioeconomic status, fewer disabled people were enrolled in the P4P program than non-disabled people (Table 2).

After further controlling for relevant variables using conditional logistic regression, the probability of enrollment in the P4P program among the disabled was found to be 0.89 times higher than that among the non-disabled (95% CI: 0.87–0.91) (Table 2). This shows that people with disabilities were less likely to enroll in the P4P program. In addition, the rate of enrollment in the P4P program was lower among patients with a higher age, catastrophic illness, higher CCI scores, and higher DCSI scores (*p* < 0.05) (Table 2); however, those with hypertension and hyperlipidemia had a higher rate of enrollment in the P4P program (*p* < 0.05).

### 3.3. Effects of Enrollment in the DM P4P Program on the Risk of Dialysis in Disabled and Non-Disabled Diabetic Patients

Table 3 shows that the proportion of dialysis occurring in both disabled and non-disabled patients was approximately 1.82%. In total, 690 (1.44%) patients who were enrolled in the DM P4P program had undergone dialysis, which is significantly lower than the proportion of those who did not enroll in the P4P program and had undergone dialysis (3345, 1.93%) (*p* < 0.05). Additionally, the Mantel–Haenszel chi-square test was used to identify any interaction effects between the presence or absence of disabilities and the P4P program participation status on the risk of dialysis; the results show that there was no interaction effect (*p* = 0.839).

Finally, the Cox PH model was used to analyze the risk of dialysis, considering the time factor, to explore the risk of dialysis from the time of diagnosis of diabetes or from the time of enrollment in the P4P program until the end of 2014. As shown in Table 3 and Figure 2, after controlling for relevant factors, the risk of dialysis was found to be significantly higher for people with disabilities than for those without disabilities (HR = 1.08, 95% CI: 1.01–1.15). Furthermore, those enrolled in the P4P program were less likely to have undergone dialysis (HR = 0.66, 95% CI: 0.60–0.72).

### 3.4. Effects of the DM P4P Program on the Mortality in Disabled and Non-Disabled Diabetic Patients

As shown in Table 3, 37.97% of the disabled patients died, which is significantly higher than the proportion of the non-disabled diabetics who died (28.02%) (*p* < 0.05). Of the 47,945 people enrolled in the P4P program, 5900 (12.3%) eventually died, significantly fewer than those (38.72%) who died and were not enrolled in the P4P program (*p* < 0.05).

As the effects of the P4P program on the care of non-disabled and disabled people may differ, the Mantel–Haenszel chi-square test revealed an interaction between the presence or absence of disability and enrollment or lack of enrollment in the P4P program in terms of the risk of death (*p* < 0.05) (Table 3). The Cox PH model was conducted by considering the interaction and the duration of diabetes, and a stratified analysis was carried out based on the presence or absence of disability as well as whether or not the patient was enrolled in the P4P program. The results are shown in Table 3 and Figure 3. In summary, based on the interaction term (Table 3), the risk of death was reduced more effectively among disabled patients compared to non-disabled patients when enrolled in the P4P program (HR = 0.87, 95% CI: 0.82–0.92). Furthermore, we conducted a stratified analysis and found that P4P care was more effective at reducing the risk of death for those with disabilities (HR = 0.41, 95% CI: 0.39–0.42) than for those without disabilities (HR = 0.47, 95% CI: 0.46–0.49) (Figure 3). Further analyses of those enrolled in the P4P program show that the risk of death of those with disabilities was significantly higher than that for those without disabilities (HR = 1.16, 95% CI: 1.10–1.22), even when enrolled in the P4P program (Figure 3).

## 4. Discussion

### 4.1. Disabled People Had a Lower Rate of Enrollment in the P4P Program than Non-Disabled People

In previous research, there was no comparison between the disabled and non-disabled in terms of their enrollment in the P4P program, nor was there exploration of their care outcomes after enrollment. In this study, the rate of enrollment in the P4P program of disabled people was significantly lower than that of non-disabled people (Table 2). According to the regulations of the P4P program in Taiwan, healthcare providers must withdraw from the program if they do not meet NHI criteria during the care process (e.g., number of follow-ups [10,11]), and this affects the ranking of the quality of care (top 25%), resulting in the loss of additional incentives. Thus, providers are inadvertently pressured to select patients who are more likely to cooperate with the P4P program. Previous studies have also shown that P4P programs tend to exclude older patients, patients with more comorbidities [36,37], or patients with more severe conditions, leading to health inequities [10,35]. Therefore, healthcare providers are less likely to include people with disabilities in the P4P program due to their poorer health statuses and the difficulty in providing them with care compared to non-disabled people [6,7,8], which may result in difficulties in achieving the relevant quality control indicators required to participate in the P4P program.

Previous studies have shown that diabetic patients’ enrollment in the DM P4P program can improve the continuity of care and reduce hospitalization utilization for diabetes-related diseases [18]; research from South Korea has shown that greater continuity of care (COC) was associated with fewer preventable hospitalizations in people with type 2 diabetes [45]. Another article adopted the definition of hospitalization from the Prevention Quality Indicator (PQI) proposed by the Agency for Healthcare Research and Quality (AHRQ) and showed that patients with type 2 diabetes who participated in the P4P program had a lower chance of preventable hospitalization [46]. Therefore, if disabled people with diabetes experience greater opportunities to enroll in the DM P4P program, similar to non-disabled people, this may help reduce preventable hospitalizations due to diabetes among people with disabilities.

### 4.2. The Similar Effects of the P4P Program on Reducing the Risk of Dialysis among DM Patients with Disabilities and Those without Disabilities

In this study, the risk of dialysis for people with disabilities was 1.08 times higher than that for people without disabilities (Table 3, Figure 2). People with disabilities have poorer health statuses [6], weaker skills in communication with healthcare providers [47,48], and more difficultly in accessing healthcare [49], resulting in poorer glycemic control and thus a higher risk of dialysis. In addition, people with disabilities may have unhealthy lifestyles, which increases the risk of weight gain and further contributes to obesity and metabolic syndromes [28,50,51,52]; these explain why diabetic patients with disabilities have a higher risk of requiring dialysis compared to non-disabled diabetic patients.

Poor glycemic control is a risk factor for diabetic nephropathy [28,50,51,52], and previous studies have shown that diabetic patients enrolled in the P4P program have more stable glycemic control [47,48]. The risk of proteinuria was reduced in those enrolled in the P4P program [47,48], as was the risk of nephropathy [28,50,51,52]. Our findings show that the risk of dialysis was lower among diabetic patients enrolled in the P4P program, regardless of whether they were disabled; their risk was significantly lower than that of those where were not enrolled (HR = 0.66, *p* < 0.05). In addition, the effectiveness of the P4P program in reducing the risk of dialysis among people with disabilities compared to that among those without disabilities when they were enrolled was not significantly different (Table 3).

### 4.3. P4P Program Enrollment Reduced the Risk of Death More for Disabled Individuals than for Non-Disabled

The ultimate goal of all healthcare is to prevent death. Table 3 shows that 12.3% of the diabetic patients who were enrolled in the P4P program died, while 38.7% of those who were not enrolled in the P4P program died. Those who were enrolled in the P4P program also had a much lower risk of dying (HR = 0.45) after controlling for relevant factors (Table 3, Figure 3). This coheres with previous studies, in which the risk of death among diabetics enrolled in the P4P program was 0.43–0.89 times lower than that of those not enrolled [19,53]. This study suggests that the reduction in the risk of death may be due to the P4P program requiring regular visits and periodic related tests. It has been revealed that diabetic patients enrolled in the P4P program have more outpatient visits where they undergo related tests for diabetes (such as HbA1c, Cr, and fundus examinations) [54], better continuity of care [19], and fewer emergency visits for infections [55], thus reducing the risk of death.

The present study has shown that the risk of death among patients with disabilities enrolled in the P4P program was significantly lower than that among patients with disabilities not enrolled in the P4P program. After controlling for other relevant factors using the Cox PH Model (Table 3, Figure 3), it was found that, among those who joined the P4P program, the risk of death for people with disabilities was 1.16 times higher than that of those without disabilities, and among those who did not participate in the P4P program, the risk of death for people with disabilities was 1.32 times higher than that of those without disabilities. The mortality rate of the physically and mentally disabled is higher than that of the non-disabled, which is consistent with the results of previous studies [7,32,33,34]. Among the disabled diabetics, the risk of death was 0.41 times lower in those enrolled in the P4P program than in those who were not, suggesting that enrollment in the P4P program is quite beneficial in mitigating the risk of death. Furthermore, P4P program enrollment reduced the risk of death to a greater extent among disabled individuals compared to non-disabled individuals, indicating that disabled individuals gain more benefits from enrolling in the P4P program. Gillani’s study found that diabetic patients with disabilities could not sufficiently self-monitor their blood glucose due to a lack of diabetes-related knowledge, but this could be effectively addressed with the assistance of medical professionals who focused more on improving patients’ knowledge and behaviors [56]. This echoes the findings of this study, wherein enrollment in the P4P program, which integrates healthcare, diet education, weight control, and the regular return monitoring of diabetic patients through the collaboration of healthcare providers and dietitians, could effectively prevent the deterioration of disabled patients due to diabetes.

## 5. Conclusions

The rates of enrollment in the P4P program were lower for people with disabilities compared to those without disabilities (OR = 0.89). Those with disabilities were at risk of rejection from the P4P program, resulting in unequal access to care. The dialysis risk was 1.08 times higher for people with disabilities than for those without disabilities, regardless of their participation in the P4P program. However, after enrollment in the DM P4P program, the mortality risk for people with disabilities was improved more than that of those without disabilities. After enrollment in the P4P program, the risk of death for people with disabilities decreased from 1.32 to 1.16 times that of persons without disabilities. For patients who enrolled in the P4P program, the risk of death was reduced significantly more for those with disabilities than for those without disabilities. Therefore, this study recommends that, in the future, the government should provide incentives to strengthen doctors’ willingness to enroll people with disabilities in the P4P program and encourage physicians to pay more attention to this disadvantaged group, thus improving the outcomes of diabetes care for people with disabilities. In future research, we suggest that studies assess the impact of P4P program enrollment on the risk of amputation and the risk of preventable hospitalization in people with disabilities and those without disabilities.

## 6. Limitations

Regarding the study’s limitations, personal health behaviors and physiological parameters such as HbA1c were not available and could not be analyzed as variables in the study. Previous studies have indicated that blood sugar control significantly affects the risk of future dialysis and death in diabetic patients. However, blood sugar data in diabetic patients was not available in this study and could not be used as a control variable for disabled and non-disabled diabetic patients. This is another limitation of this study.

## Figures and Tables

**Figure 1 healthcare-11-02742-f001:**
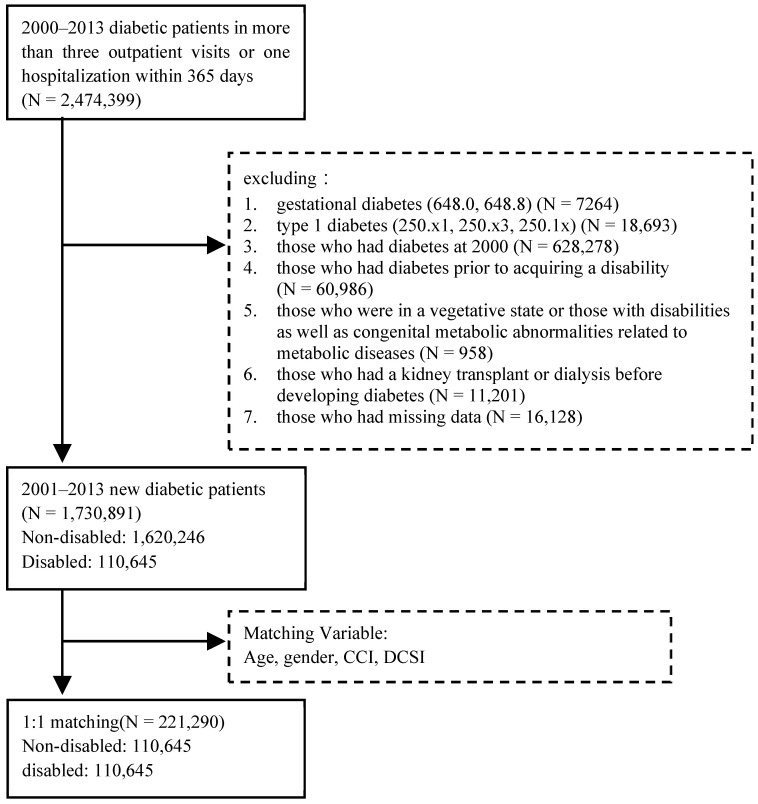
Screening process for the study participants.

**Figure 2 healthcare-11-02742-f002:**
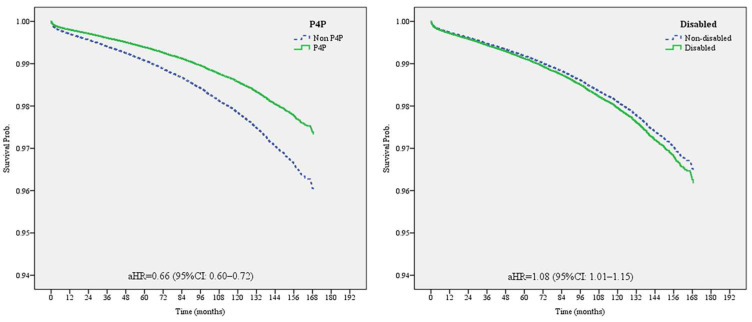
Comparing the effects of P4P program participation status and disability status on the dialysis risk in type 2 diabetic patients via a Cox proportional hazard model (after controlling for sex, age, monthly salary, urbanization of residence area, CCI, DCSI, hypertension, hyperlipidemia, physician volume, healthcare organization level, and healthcare organization type).

**Figure 3 healthcare-11-02742-f003:**
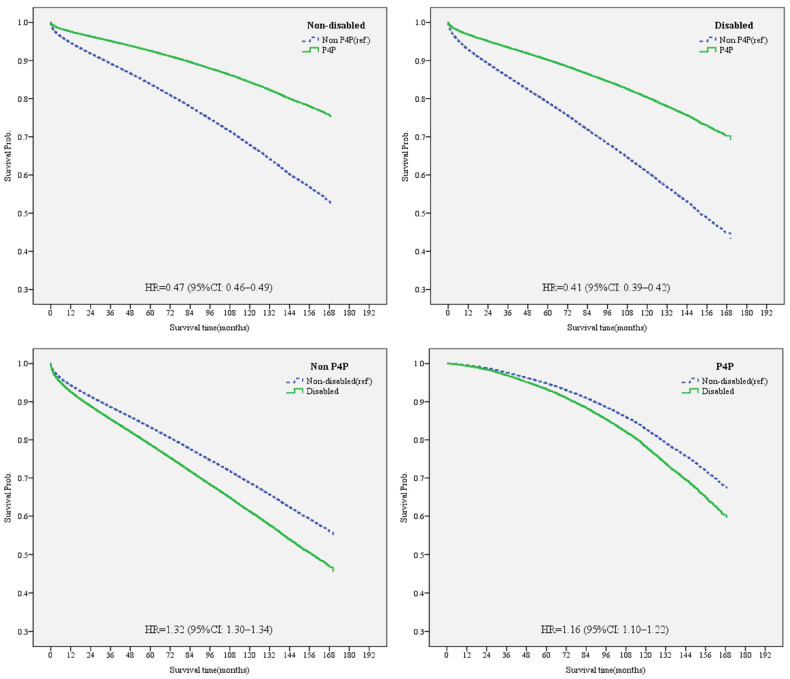
Stratified analysis: comparing the effects of P4P program participation status and disability status on the risk of death in diabetic patients via a Cox proportional hazard model (after controlling for sex, age, monthly salary, urbanization of residence area, CCI, DCSI, hypertension, hyperlipidemia, physician volume, healthcare organization level, and healthcare organization type).

**Table 1 healthcare-11-02742-t001:** Bivariate analysis of type 2 diabetes patients with and without a disability before and after matching.

		Before Matching						SMD *	After 1:1 Matching						SMD
		Total (N = 1,730,891)		Non-Disabled (*n* = 1,620,246)		Disabled (*n* = 110,645)			Total (N = 221,290)		Non-Disabled (*n* = 110,645)		Disabled (*n* = 110,645)		
		N	%	*n*	%	*n*	%		N	%	*n*	%	*n*	%	
Sex															
	Female	803,979	46.45	755,536	46.63	48,443	43.78	−0.057	96,886	43.78	48,443	43.78	48,443	43.78	0.000
	Male	926,912	53.55	864,710	53.37	62,202	56.22	0.057	124,404	56.22	62,202	56.22	62,202	56.22	0.000
Age (years)															
	<45	289,012	16.70	275,097	16.98	13,915	12.58	−0.124	27,826	12.57	13,911	12.57	13,915	12.58	0.000
	45~54	448,953	25.94	427,542	26.39	21,411	19.35	−0.168	42,825	19.35	21,414	19.35	21,411	19.35	0.000
	55~64	438,980	25.36	416,713	25.72	22,267	20.12	−0.134	44,534	20.12	22,267	20.12	22,267	20.12	0.000
	65~74	325,298	18.79	301,210	18.59	24,088	21.77	0.079	48,176	21.77	24,088	21.77	24,088	21.77	0.000
	≥75	228,648	13.21	199,684	12.32	28,964	26.18	0.357	57,929	26.18	28,965	26.18	28,964	26.18	0.000
Average age (SD)		58.26 (14.15)		57.92 (13.98)		63.27 (15.57)		0.897	62.24 (15.29)		62.10 (15.09)		63.27 (15.57)		0.102
CCI															
	0	911,154	52.64	872,021	53.82	39,133	35.37	−0.378	78,266	35.37	39,133	35.37	39,133	35.37	0.000
	1	428,741	24.77	401,390	24.77	27,351	24.72	−0.001	54,702	24.72	27,351	24.72	27,351	24.72	0.000
	2	197,119	11.39	179,242	11.06	17,877	16.16	0.149	35,754	16.16	17,877	16.16	17,877	16.16	0.000
	≥3	193,877	11.20	167,593	10.34	26,284	23.76	0.363	52,568	23.76	26,284	23.76	26,284	23.76	0.000
DCSI															
	0	1,390,678	80.34	1,316,387	81.25	74,291	67.14	−0.327	148,583	67.14	74,292	67.14	74,291	67.14	0.000
	1	170,318	9.84	159,034	9.82	11,284	10.20	0.013	22,568	10.20	11,284	10.20	11,284	10.20	0.000
	2	132,930	7.68	114,382	7.06	18,548	16.76	0.303	37,096	16.76	18,548	16.76	18,548	16.76	0.000
	≥3	36,965	2.14	30,443	1.88	6522	5.89	0.209	13,043	5.89	6521	5.89	6522	5.89	0.000

* SMD: standardized mean difference.

**Table 2 healthcare-11-02742-t002:** Bivariate analysis and logistic regression analysis of variables in type 2 diabetes patients with or without a disability and P4P program participation status.

		Total	Non-Disabled		Disabled		Non-Disabled				Disabled				P4P Program Participation			
			Total		Total		Non P4P		P4P		Non P4P		P4P					
			N	%	N	%	*n*	%	*n*	%	*n*	%	*n*	%	aOR	95% CI		*p* Value
Total		221,290	110,645	100.00	110,645	100.00	84,954	76.78	25,691	23.22	88,391	79.89	22,254	20.11				
	Non-disabled	110,645	110,645	100.00	-	-	-	-	-	-	-	-	-	-	1.00			
	Disabled	110,645	-	-	110,645	100.00	-	-	-	-	-	-	-	-	0.89	0.87	0.91	<0.001
P4P																		
	Non-P4P	173,345	84,954	76.78	88,391	79.89	84,954	76.78	25,691	23.22	-	-	-	-				
	P4P	47,945	25,691	23.22	22,254	20.11	-	-	-	-	88,391	79.89	22,254	20.11				
Sex																		
	Female	96,886	48,443	43.78	48,443	43.78	37,349	77.10	11,094	22.90	38,866	80.23	9577	19.77	1.00			
	Male	124,404	62,202	56.22	62,202	56.22	47,605	76.53	14,597	23.47	49,525	79.62	12,677	20.38	1.07	1.05	1.1	<0.001
Age (years)																		
	<45	27,826	13,911	12.57	13,915	12.58	9528	68.49	4383	31.51	9795	70.39	4120	29.61	1.00			
	45~54	42,825	21,414	19.35	21,411	19.35	14,913	69.64	6501	30.36	15,205	71.01	6206	28.99	0.92	0.89	0.95	<0.001
	55~64	44,534	22,267	20.12	22,267	20.12	15,989	71.81	6278	28.19	16,657	74.81	5610	25.19	0.81	0.78	0.84	<0.001
	65~74	48,176	24,088	21.77	24,088	21.77	18,652	77.43	5436	22.57	19,779	82.11	4309	17.89	0.74	0.71	0.77	<0.001
	≥75	57,929	28,965	26.18	28,964	26.18	25,872	89.32	3093	10.68	26,955	93.06	2009	6.94	0.43	0.41	0.46	<0.001
Average age (SD)		63.11 (15.3)	62.95 (15.09)		63.27 (15.57)		64.44 (15.25)		58.03 (13.43)		64.92 (15.65)		56.71 (13.38)					
Monthly salary (TWD)																		
	≤17,280	15,874	5307	4.8	10,567	9.55	4109	77.43	1198	22.57	8486	80.31	2081	19.69	1.00			
	17,281~28,800	149,309	72,199	65.25	77,110	69.69	56,284	77.96	15,915	22.04	62,041	80.46	15,069	19.54	1.12	1.08	1.17	<0.001
	28,801~45,800	35,743	20,432	18.47	15,311	13.84	14,844	72.65	5588	27.35	11,557	75.48	3754	24.52	1.21	1.15	1.27	<0.001
	45,801~57,800	6787	4152	3.75	2635	2.38	3099	74.64	1053	25.36	2135	81.02	500	18.98	1.12	1.04	1.2	0.004
	≥57,801	13,442	8504	7.69	4938	4.46	6581	77.39	1923	22.61	4107	83.17	831	16.83	1.02	0.96	1.08	0.605
	Missing	135	51	0.05	84	0.08												
Urbanization of residence area																		
	Level 1	51,731	30,061	27.17	21,670	19.59	23,437	77.96	6624	22.04	17,690	81.63	3980	18.37	1.00			
	Level 2	62,011	32,674	29.53	29,337	26.51	24,822	75.97	7852	24.03	22,902	78.07	6435	21.93	1.19	1.15	1.22	<0.001
	Level 3	34,346	17,565	15.88	16,781	15.17	13,482	76.75	4083	23.25	13,367	79.66	3414	20.34	1.17	1.13	1.21	<0.001
	Level 4	38,983	17,010	15.37	21,973	19.86	12,788	75.18	4222	24.82	17,501	79.65	4472	20.35	1.36	1.31	1.41	<0.001
	Level 5	7981	3251	2.94	4730	4.27	2613	80.38	638	19.62	3887	82.18	843	17.82	1.14	1.07	1.22	<0.001
	Level 6	14,344	5603	5.06	8741	7.90	4390	78.35	1213	21.65	6950	79.51	1791	20.49	1.28	1.22	1.35	<0.001
	Level 7	11,894	4481	4.05	7413	6.700	3422	76.37	1059	23.63	6094	82.21	1319	17.79	1.21	1.14	1.28	<0.001
Catastrophic illness																		
	No	153,669	85,478	77.25	68,191	61.63	63,988	74.86	21,490	25.14	53,093	77.86	15,098	22.14	1.00			
	Yes	67,621	25,167	22.75	42,454	38.37	20,966	83.31	4201	16.69	35,298	83.14	7156	16.86	0.87	0.85	0.9	<0.001
CCI																		
	0	78,266	39,133	35.37	39,133	35.37	27,541	70.38	11,592	29.62	28,479	72.77	10,654	27.23	1.00			
	1	54,702	27,351	24.72	27,351	24.72	20,694	75.66	6657	24.34	21,566	78.85	5785	21.15	1.08	1.04	1.13	<0.001
	2	35,754	17,877	16.16	17,877	16.16	14,245	79.68	3632	20.32	14,892	83.30	2985	16.70	1.13	1.07	1.19	<0.001
	≥3	52,568	26,284	23.76	26,284	23.76	22,474	85.50	3810	14.50	23,454	89.23	2830	10.77	1.01	0.95	1.07	0.875
DCSI																		
	0	148,583	74,292	67.14	74,291	67.14	55,188	74.29	19,104	25.71	57,123	76.89	17,168	23.11	1.00			
	1	22,568	11,284	10.2	11,284	10.2	8454	74.92	2830	25.08	8871	78.62	2413	21.38	1.05	1.01	1.09	0.009
	2	37,096	18,548	16.76	18,548	16.76	15,652	84.39	2896	15.61	16,470	88.80	2078	11.20	0.92	0.88	0.97	<0.001
	≥3	13,043	6521	5.89	6522	5.89	5660	86.80	861	13.20	5927	90.88	595	9.12	0.85	0.79	0.92	<0.001
Hypertension																		
	No	44,134	22,603	20.43	21,531	19.46	17,385	76.91	5218	23.09	17,135	79.58	4396	20.42	1.00			
	Yes	177,156	88,042	79.57	89,114	80.54	67,569	76.75	20,473	23.25	71,256	79.96	17,858	20.04	1.21	1.18	1.25	<0.001
Hyperlipidemia																		
	No	86,266	37,485	33.88	48,781	44.09	32,923	87.83	4562	12.17	44,085	90.37	4696	9.63	1.00			
	Yes	135,024	73,160	66.12	61,864	55.91	52,031	71.12	21,129	28.88	44,306	71.62	17,558	28.38	2.65	2.58	2.72	<0.001
Physician volume																		
	Low (<Q1)	9027	4401	3.98	4626	4.18	3280	74.53	1121	25.47	3579	77.37	1047	22.63	1.00			
	Median (Q3~Q1)	60,701	29,614	26.76	31,087	28.1	23,120	78.07	6494	21.93	25,078	80.67	6009	19.33	1.00	0.95	1.06	0.977
	High (>Q3)	151,562	76,630	69.26	74,932	67.72	58,554	76.41	18,076	23.59	59,734	79.72	15,198	20.28	1.09	1.03	1.15	0.002
Healthcare organization level																		
	Medical center	44,734	23,619	21.35	21,115	19.08	18,983	80.37	4636	19.63	17,177	81.35	3938	18.65	1.00			
	Regional hospital	67,861	32,044	28.96	35,817	32.37	24,019	74.96	8025	25.04	27,963	78.07	7854	21.93	1.31	1.27	1.35	<0.001
	District hospital	51,314	23,008	20.79	28,306	25.58	18,121	78.76	4887	21.24	23,688	83.69	4618	16.31	1.01	0.98	1.05	0.540
	Community clinic	52,800	30,408	27.48	22,392	20.24	22,553	74.17	7855	25.83	16,934	75.63	5458	24.37	0.90	0.87	0.93	<0.001
	Missing	4581	1566	1.42	3015	2.72												
Healthcare organization type																		
	Public	8715	4805	4.34	3910	3.53	3264	67.93	1541	32.07	2630	67.26	1280	32.74	1.00			
	Non-public	212,575	105,840	95.66	106,735	96.47	81,690	77.18	24,150	22.82	85,761	80.35	20,974	19.65	0.57	0.54	0.61	<0.001

**Table 3 healthcare-11-02742-t003:** The effects of disability status and P4P participation status on the risks of dialysis and death.

		**Total**		**Non Dialysis**		**Dialysis**		***p* Value**	**Risk of Dialysis ^#^**			
		**N**	**%**	** *n* **	**%**	** *n* **	**%**		**aHR**	**95% CI**		***p* Value**
Total		221,290	100.00	217,255	98.18	4035	1.82					
Status								0.898				
	Non-disabled	110,645	50.00	108,632	98.18	2013	1.82		1.00			
	Disabled	110,645	50.00	108,623	98.17	2022	1.83		1.08	1.01	1.15	0.026
P4P								<0.001				
	Non P4P	173,345	78.33	170,000	98.07	3345	1.93		1.00			
	P4P	47,945	21.67	25,323	52.82	690	1.44		0.66	0.6	0.72	<0.001
Status × P4P interaction term								0.839				
		**Total**		**Survival**		**Death**		***p* Value**	**Risk of Death ^#^**			
		**N**	**%**	** *n* **	**%**	** *n* **	**%**		**aHR**	**95% CI**		***p* Value**
Total		221,290	100.00	148,275	67.00	73,015	33.00					
Status								<0.001				
	Non-disabled	110,645	50.00	79,645	71.98	31,000	28.02		1.00			
	Disabled	110,645	50.00	68,630	62.03	42,015	37.97		1.34	1.32	1.36	<0.001
P4P								<0.001				
	Non P4P	173,345	78.33	106,230	61.28	67,115	38.72		1.00			
	P4P	47,945	21.67	42,045	87.69	5900	12.31		0.45	0.43	0.47	<0.001
Status × P4P interaction term								<0.001	0.87	0.82	0.92	<0.001

^#^ All models have been controlled for sex, age, monthly salary, urbanization of residence area, CCI, DCSI, hypertension, hyperlipidemia, physician volume, healthcare organization level, and healthcare organization type. aHR: adjusted hazard ratio.

## Data Availability

Data are available from the Health and Welfare Data Science Center of the Ministry of Health and Welfare (MOHW) (https://www.mohw.gov.tw/mp-2.html (accessed on 1 May 2023)), Taiwan. All interested researchers can apply to use the database managed by the MOHW. Due to legal restrictions imposed by the Taiwanese government related to the Personal Information Protection Act, the database cannot be made publicly available. Raw data from the Health and Welfare Data Science Center cannot be brought out. The restrictions prohibited the authors from making the basic data set publicly available.

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
