# Peer review of "Participation and Outcomes among Disabled and Non-Disabled People in the Diabetes Pay-for-Performance Program"

_healthcare, 2023, doi:10.3390/healthcare11202742_

Round 1
Reviewer 1 Report
See attachment

Reviewer 2 Report
Dear Authors,
We recently had the opportunity to read your manuscript titled “Participation and Outcomes between Disabled and Non-disabled People in Pay-for-Performance”, and we wanted to reach out to you to express our comments about your work.
Your study used national databases in Taiwan to compare participation in a diabetes pay-for-performance (P4P) program and outcomes between disabled and non-disabled patients newly diagnosed with diabetes from 2001-2013. After propensity score matching, disabled patients were less likely to enroll in the P4P program but had higher risks of dialysis and death. P4P enrollment significantly reduced mortality, especially for disabled patients. Despite worse outcomes overall, P4P benefits on reducing mortality were greater for disabled versus non-disabled diabetic patients.
Nevertheless, here are some possible comments outlining areas that could improve the quality and readability of the manuscript:
Introduction:
1. Aims and objectives could be more clearly and explicitly stated early on.
2. Background provides good context but could expand on prior research on P4P programs.
3. More discussion of research gap and rationale for studying disabled population specifically
Methods:
4. Participant disability criteria explained clearly.
5. Description of data sources and variables is thorough.
6. Propensity score matching method appropriate but details lacking.
7. Statistical analysis and the regression models: The authors use appropriate regression techniques like conditional logistic regression and Cox proportional hazards models. However, model building and variable selection methods are not described. Were interaction terms properly specified and interpreted? Were underlying assumptions tested and met?
8. Overall statistical approach seems sound but lacks some specifics that would give confidence the analysis was performed correctly. How was missing data handled? Were sensitivity analyses done?
9. The 1:1 matching process is not clearly detailed. It could be adding several bias in the selection of final Non-disabled participants for reaching the 1:1 parity.
Results:
10. Key findings align with aims though organization could highlight objectives better.
11. Some tables have redundancies and could be consolidated.
12. Results section is dense - visualizations and headers could improve clarity.
Discussion:
13. Summarizes key findings but language is technical - rewrite for broad audience.
14. Limitations around unavailable data acknowledged.
15. Could expand on implications for policy and future research.
Conclusions:
16. Research aims are addressed but conclusion is brief. Please, modify it.
17. Significance and implications could be emphasized more.
18. Practical recommendations for stakeholders are not included. Please, add them.
Regarding the grammar, while generally well-written, the manuscript requires extensive language editing to meet publishing standards. Numerous grammatical errors need to be corrected. Sentence structure could be tightened by breaking up lengthy, complex sentences and using more concise phrasing. Attention should be paid to improving clarity through paragraph structure, headers, and flow between ideas. Terminology needs to be defined, used consistently, and checked against a style guide. Tense inconsistency, citation formatting errors, spelling inconsistencies, and section header hierarchy issues also need to be rectified. Robust editing by a native English speaker familiar with scientific writing conventions is imperative to improve readability, clarity, grammar, and adherence to publishing standards expected for a top journal as Healthcare is.
Once again, thank you very much for your work. We´ll be waiting for your answers about our comments.
Kindest regards,
Regarding the grammar, while generally well-written, the manuscript requires extensive language editing to meet publishing standards. Numerous grammatical errors need to be corrected. Sentence structure could be tightened by breaking up lengthy, complex sentences and using more concise phrasing. Attention should be paid to improving clarity through paragraph structure, headers, and flow between ideas. Terminology needs to be defined, used consistently, and checked against a style guide. Tense inconsistency, citation formatting errors, spelling inconsistencies, and section header hierarchy issues also need to be rectified. Robust editing by a native English speaker familiar with scientific writing conventions is imperative to improve readability, clarity, grammar, and adherence to publishing standards expected for a top journal as Healthcare is.
Reviewer 3 Report
I was invited to revise the paper entitled "Participation and Outcomes between Disabled and Nondisabled People in Pay-for-Performance". It aimed to evaluate outcomes of Taiwanese diabetic patients enrolled in a pay for performance program known as DM P4P.
The topic is interesting and faced off the rising problem of diabetes care in the South-East.
Observations:
- The title should be more informative. The term "Diabetes" should be added;
- Methodology is strong and appropriate. I suggesto to add in table 1 the standardized mean difference between disabled and non disabled after matching procedure;
- Did Authors have information about long term diabetes complications such as amputations? If possible, I suggest to perform a cox model evaluating the amputation outcome;
- In discussion section, Authors should evaluate the impact of possible diabetes preventable hospitalizations, as described by AHRQ. Several studies evaluated these outcomes from other countries, Also, Authors should compare their results with previous literature from other countries.
Round 2
Reviewer 2 Report
Dear Authors,
We recently had the opportunity to read your modified version of the manuscript titled “Participation and Outcomes between Disabled and Non-disabled People in Pay-for-Performance”, and we wanted to reach out to you to express our comments about your work.
The reviewers have been reading your responses to our comments but have found that most of them make no reference to the modifications made, if any, nor do many of your responses indicate where to find these modifications. Reviewers cannot search the entire text for every modification and the response used to most of our comments has been "Thanks for the reviewer's comment", without any additional explanation or commentary to provide more information, which is something very unsual. It is true that on other issues some information has been provided but we cannot do a proper review without all the information needed to do so. Therefore, we ask you to respond to our previous comments in a professional manner, detailing the modifications made, if any, and their location in the text.
We look forward to hearing from you and your feedback about our comments.
Once again, thank you very much for your work. We´ll be waiting for your answers about our comments.
Kindest regards,
The manuscript needs extensive language editing to meet publishing standards. It contains numerous grammatical errors, long and complex sentences, and issues with clarity. To improve readability and adherence to publishing standards, it's essential to address terminology consistency, tense errors, citation formatting, spelling, and section headers. Comprehensive editing by a native English speaker familiar with scientific writing conventions is crucial for publication in a top journal like Healthcare.
Reviewer 3 Report
Authors addressed points 1 and 2. About points 3, the lack of this information and the unavailability of it should be added in limitation section. About point 4, authors should discuss about preventable hospitalizations, as stated in the previous revision round.
